# Comparative Research: Regulatory Mechanisms of Ribosomal Gene Transcription in *Saccharomyces cerevisiae* and *Schizosaccharomyces pombe*

**DOI:** 10.3390/biom13020288

**Published:** 2023-02-03

**Authors:** Hayato Hirai, Kunihiro Ohta

**Affiliations:** 1Department of Life Sciences, Graduate School of Arts and Sciences, The University of Tokyo, Komaba 3-8-1, Tokyo 153-8902, Japan; 2Universal Biology Institute, The University of Tokyo, Hongo 7-3-1, Tokyo 113-0033, Japan

**Keywords:** budding yeast, fission yeast, ribosome, heterochromatin, epigenetics, gene regulation, TORC1, TOR pathway

## Abstract

Restricting ribosome biosynthesis and assembly in response to nutrient starvation is a universal phenomenon that enables cells to survive with limited intracellular resources. When cells experience starvation, nutrient signaling pathways, such as the target of rapamycin (TOR) and protein kinase A (PKA), become quiescent, leading to several transcription factors and histone modification enzymes cooperatively and rapidly repressing ribosomal genes. Fission yeast has factors for heterochromatin formation similar to mammalian cells, such as H3K9 methyltransferase and HP1 protein, which are absent in budding yeast. However, limited studies on heterochromatinization in ribosomal genes have been conducted on fission yeast. Herein, we shed light on and compare the regulatory mechanisms of ribosomal gene transcription in two species with the latest insights.

## 1. Introduction

The ribosome is massive intracellular machinery translating mRNA, transcribed from DNA, into proteins in living organisms. This machinery is constructed by a small 40S subunit composed of 18S ribosomal RNA (rRNA) and 33 ribosomal proteins (RPs) and a large 60S subunit composed of three rRNAs (5S, 5.8S, and 25/28S) and 46 (in yeast) or 47 (in human) RPs [1]. It has been estimated that 2,000 ribosomes are synthesized per minute in the budding yeast *Saccharomyces cerevisiae*, and each cell contains 100,000–200,000 ribosomes in the fission yeast *Schizosaccharomyces pombe* [2,3]. To sustain large amounts of ribosomes, rRNA and RP genes must be vigorously transcribed. Indeed, rRNA accounts for about 80% of intracellular RNA in *S. cerevisiae* [2], and RP genes are highly transcribed by RNA polymerase II in both *S. cerevisiae* and *S. pombe* [3,4].

To achieve rapid and robust rRNA transcription, multiple copies of ribosomal DNA (rDNA) containing 18S, 5.8S, and 25S/28S regions exist tandemly in the genome (Figure 1). For example, *S. cerevisiae* carries approximately 150 rDNA tandem repeats on chromosome XII, *S. pombe* has 100–150 repeats on both ends of chromosome III, and humans carry a total of ~350 repeats on five different chromosomes [5,6,7]. The 5S rDNA gene is encoded on the same repeat of chromosome XII in *S. cerevisiae* [5], whereas multiple 5S rDNA genes are dispersed on three chromosomes in *S. pombe* [8]. About 100 repeats of 5S rDNA genes exist in humans on chromosome I [9]. Furthermore, some RP genes are found as paralogs due to genome duplication in *S. cerevisiae* [10].

Transcription of ribosome-associated genes requires three RNA polymerases. That is, consecutive sequences of 18S, 5.8S, and 25S/28S rRNA are transcribed by RNA polymerase I in the nucleolus as the precursor rRNA from 5′ETS (external transcribed spacer) to 3′ETS, including ITS (internal transcribed spacer) 1 and ITS 2 (Figure 1). 5S rRNA is transcribed by RNA polymerase III, and the RP genes are transcribed by RNA polymerase II [11]. Both small and large ribosomal subunit assemblies proceed in parallel with the removal of four transcribed spacers in the precursor rRNA by endonuclease and exonuclease processing [1,11,12]. These reactions require more than 200 assembly factors (Ribi) and 77 snoRNAs in *S. cerevisiae* and more than 500 assembly factors and 300 snoRNAs in humans [1].

In summary, numerous factors contribute to ribosome biogenesis, and these genes are actively transcribed. Accordingly, intracellular resources are substantially devoted to ribosome synthesis. Therefore, ribosome biosynthesis must be promptly halted to reduce intracellular energy consumption when the cells encounter nutrient depletion.

One way to cease ribosome biogenesis is epigenetic gene regulation through modifying histones and methylation of cytosine residues on DNA without altering the DNA sequences. DNA methylation does not occur in budding and fission yeasts [13,14]. Histone modifications, therefore, play a pivotal role in gene silencing. In fission yeast, the silent state of chromatin is accomplished by the methylation of histone H3 lysine 9 (H3K9me), followed by the recruitment of heterochromatin protein 1 (HP1) [15,16]. Recently, regions of heterochromatinization in response to environmental changes (namely, facultative heterochromatin) have been identified [17,18]. Some ribosomal genes are suppressed by heterochromatin formation in response to starvation [17]; however, the mechanisms of facultative heterochromatin formation have not been fully unveiled. Crucially, unlike fission yeast, there is no H3K9 methyltransferase Clr4 (SUV39H1 in humans) and HP1 in budding yeast [19]. Therefore, the mechanism of transcriptional repression is fundamentally distinct between budding and fission yeasts.

This review summarizes the findings on the transcriptional regulation mechanisms of ribosomal genes and compares the difference between budding and fission yeasts.

## 2. Transcription of the Ribosomal Genes in the Stress Response Is Regulated by the TOR and PKA Pathways

When the nutrient supply decreases, ribosome biosynthesis must be halted immediately to avoid the consumption of intracellular resources. Early studies using *S. cerevisiae* revealed that the concentration of active rDNA is reduced in the stationary phase (or when cells are cultured in a minimal medium) compared to that in the logarithmic growth phase (or when cells are cultured in a nutrient-rich medium) [20]. In addition, the expression of RP genes was repressed in the presence of 3-amino-1,2,4-triazole (3-AT), a metabolic antagonist that inhibits histidine synthesis [21], and tyrosine depletion reduced both rRNA and RP gene mRNA levels [22], indicating that amino acid starvation attenuates the transcription of ribosome-related genes. Subsequently, microarray analysis provided ample evidence that RP genes are repressed due to environmental stresses, including nitrogen and glucose starvation [23]. Jorgensen and colleagues also showed that the expression of RP and Ribi genes decreases 30 min after glucose depletion [24].

In *S. pombe*, the depletion of sulfur, an indispensable component for many coenzymes such as coenzyme A and biotin, diminishes the expression of ribosome-related genes [25]. A comprehensive analysis using RNA-seq indicated that nitrogen starvation reduces rRNA transcription levels by about 1/10 [3]. We also revealed that a group of gene expressions associated with ribosome biosynthesis was decreased within 30 min after glucose starvation [26,27].

Intriguingly, ribosomal genes can also be repressed by stresses other than nutritional starvation. For example, mild heat shock leads to quick but transient repression of RP genes in *S. cerevisiae* [22,28]. In *S. pombe*, oxidative, cadmium, heat, osmotic, and DNA damage stress commonly repress ribosomal genes [29]. Since nutrient signaling pathways, such as the target of rapamycin (TOR) and protein kinase A (PKA), are intensely involved in regulating ribosomal genes, it is, therefore, interesting to study how stresses, unlike starvation, can repress these genes.

TOR was initially identified as a target gene for the macrolide immunosuppressant drug rapamycin by genetic screening using *S. cerevisiae*. Two types of TOR proteins, Tor1 and Tor2, are found in budding yeast, forming TOR complex 1 (TORC1) and TOR complex 2 (TORC2), respectively [30,31]. Notably, starved and rapamycin-treated cells showed a similar phenotype, suggesting that the TOR signaling pathway recognizes nutrients and promotes cell proliferation [32,33]. With this background, the suppression of ribosome-associated genes by starvation was speculated to be due to TOR inactivation. Indeed, rapamycin’s inhibition of the TOR pathway rapidly downregulates the transcription of both rRNA and RP genes [34,35,36] (Figure 2). In fission yeast, TORC1 (composed of Tor2 protein) and TORC2 (of Tor1 protein) were identified, and TORC1 mainly regulates the expression of ribosome-related genes [37,38,39].

The PKA pathway regulates most glucose-dependent transcriptional responses [40]. PKA forms a heterotetramer consisting of two regulatory subunits of Bcy1 and two catalytic subunits of either of the three isoforms Tpk1/2/3. In the presence of glucose, the GTP-binding proteins Ras1/2 and Gpa2 activate the adenylate cyclase Cyr1, which converts ATP to cyclic AMP (cAMP), resulting in the inhibition of Bcy1, and free Tpk is activated [40,41,42,43] (Figure 2). In *S. cerevisiae*, the external addition of cAMP in a mutant capable of its uptake upregulates the expression of RP genes, and the depletion of cAMP downregulates their expression [44]. Furthermore, a comprehensive analysis using DNA microarrays showed that PKA and TOR cooperate in the transcriptional changes in ribosomal genes in response to fluctuating glucose levels [45].

Meanwhile, in *S. pombe*, the PKA pathway regulates the expression of some genes, including *fbp1* and *ste11* [46,47]. In contrast, there is no evidence that the pathway regulates the expression of ribosome-related genes. Therefore, a comprehensive expression analysis by RNA-seq using mutants that lack a functional PKA pathway could be highly informative.

Taken together, the TOR and PKA pathways, at least in *S. cerevisiae*, primarily regulate the expression of ribosome-associated genes by sensing external nutritional conditions. In the next section, we address specific mechanisms by which TOR and PKA regulate gene expression in budding yeast.

## 3. Regulatory Mechanisms of Ribosome-Related Gene Expression in *S. cerevisiae*

Ribosome-related genes can be classified into three categories: rRNA, RP genes, and Ribi genes. A universal pathway does not regulate these categories of genes; therefore, we present a summary of what is currently known about the regulatory mechanisms in each of these groups.

### 3.1. Regulatory Mechanisms of rRNA Expression

A region with the rDNA gene comprising approximately 9.1 kb of transcribed and non-transcribed spacers is repeated 150 times on chromosome XII in *S. cerevisiae* [5]. Analysis using a chromatin spread visualized by electron microscopy developed by Oscar Miller [48] suggests that active transcription only occurs in 50% of these rDNA repeats [20]. This analysis inferred that open and closed chromatin states couple with active and inactive transcription, respectively. The dynamics of chromatin structure are supposed to be defined by the distribution and modifications of its histone proteins. Staphylococcal nuclease and DNase I digestion assays implied that inactivated rDNA displays a homogenous chromatin pattern [49]. A study investigating the H3 and H2B profile using chromatin immunoprecipitation (ChIP) found that they are allocated throughout the 35S rDNA repeats. However, this strategy does not distinguish whether histones are homogenously distributed in both active and inactive rDNA. Notably, a mutant with fewer rDNA repeats showed activation of almost all rDNA accompanied by a reduction in histones on rDNA and a precarious chromatin structure [50,51]. These data suggest irregular and unstable chromatin structures are formed on actively transcribed rDNA units with low histone density (Figure 3).

Instead of histones, the HMG-box protein Hmo1 (which shares its function with linker histone H1) strongly accumulates on 35S rDNA through the DNA binding domain box A and box B, forming a fragile chromatin structure [52,53,54,55,56]. During DNA synthesis (S phase), nucleosomes fill the rDNA regions. However, as the cell cycle progresses, RNA polymerase I transcription displaces the nucleosome and facilitates the loading of Hmo1 into rDNA, thereby establishing open chromatin [57]. Since 18S and 25S rRNA transcript levels are reduced in *hmo1*∆ cells, nucleosome repackaging into rDNA may repress transcription [52,58]. An investigation of the Hmo1 behavior after rapamycin treatment mimicking nutrient starvation showed that it rapidly dissociates from 35S rDNA [53]. These findings suggest that TORC1 inactivation represses rRNA transcription by dissociating Hmo1 from rDNA. The level of Hmo1 expression may be able to explain this dissociation. Tor1 and Hmo1 localize on the *HMO1* gene promoter in a nutrient medium. In contrast, they dissociate upon rapamycin treatment, leading to the recruitment of the repressor Crf1 to the promoter repression of the *HMO1* gene [59,60] (Figure 3).

In summary, inhibiting canonical nucleosome formation by Hmo1 activates rRNA transcription. Upon starvation, TORC1 inactivation reduces the expression of the *HMO1* gene, causing dissociation of Hmo1 and accumulation of nucleosomes on the rDNA, ultimately leading to the silencing of rRNA transcription (Figure 3). However, the dissociation of Hmo1 from rDNA upon starvation may not be sufficient to repress rRNA transcription rapidly.

The degradation of RNA polymerase I in response to starvation may be another mechanism for *S. cerevisiae* rRNA repression. RNA polymerase I, in concert with core factor CF [61], upstream activation factor UAF [62], TATA-binding protein TBP [63], and monomeric factor Rrn3 [64], are recruited to the rDNA promoter. Among them, rapamycin treatment diminishes *RRN3* transcription and causes degradation of pre-existing Rrn3 [65]. Consequently, the complex of Rrn3 with RNA polymerase I is disrupted in the nucleolus, and rRNA transcription is attenuated [66] (Figure 3). Moreover, phosphorylation by TOR may promote the association of Rrn3 with RNA polymerase I since dephosphorylated RNA polymerase I dissociates with Rrn3 in vitro [67]. Artificial fusion of Rrn3 to RNA polymerase I subunit Rpa43 alleviates the reduction in rRNA transcription, albeit partially caused by nutrient starvation and rapamycin treatment, indicating that multilayered pathways contribute to rRNA repression [68].

In *S. cerevisiae*, the serine/threonine kinase TORC1 primarily localizes to the vacuolar membrane [69] and phosphorylates substrates, such as the homolog of mammalian S6 kinase, Sch9 [70]. If true, how can TORC1, located in the cytosol, swiftly regulate rRNA expression? Remarkably, Tor1 carries two nuclear localization signal (NLS) sequences and one nuclear export signal (NES) sequence and binds to the promoter of 35S rDNA and the entire 5S rDNA through its helix–turn–helix (HTH) domain in the presence of ambient nutrients [71,72]. Thus, TORC1 on rDNA might directly phosphorylate and activate Hmo1 and Rrn3.

In addition to these factors, the *S. cerevisiae* transcriptional repressor Maf1 is phosphorylated by Sch9, the downstream target of the TORC1 pathway, resulting in its dissociation from 5S rDNA, where transcription by RNA polymerase III is activated. Conversely, upon starvation or rapamycin treatment, dephosphorylated Maf1 quenches RNA polymerase III-dependent transcription as a negative cofactor [72,73,74,75].

Finally, we refer to a case where rDNA transcription is epigenetically regulated in *S. cerevisiae*. Crucially, the H3K9 methyltransferase SUV39H1 and heterochromatin protein HP1, essential for heterochromatin formation, are missing in budding yeast [19]. Hence, the histone deacetylase (HDAC) Sir2 and Rpd3 are supposed to function in chromatin condensation and silencing in *S. cerevisiae* rDNA. Although Rpd3, but not Sir2, accumulates in rDNA after the inactivation of TORC1 via rapamycin treatment and facilitates a closed chromatin formation in rDNA repeats (Figure 3), it does not contribute to the repression of rRNA production itself [76,77]. *S. cerevisiae* Sir2 prevents mitotic recombination by repressing transcription from the non-coding bidirectional promoter (E-pro) in the rDNA region rather than by repressing transcription of 35S rRNA [78]. Interestingly, in mammals, the deacetylation of the RNA polymerase I subunit PAF53 (A49 in *S. cerevisiae*) by SIRT7 (Sir2 in S. *cerevisiae*) enhances the association of histone and RNA polymerase I on rDNA, leading to an upregulation of rRNA transcription [79,80].

Instead, the *S. cerevisiae* COMPASS complex involving Set1 promotes histone H3K4 methylation and represses transcription from rDNA during nutrient-rich conditions, suggesting that the Set1 complex silences the dormant rDNA unit [81,82] (Figure 3). However, whether Set1 also contributes to *S. cerevisiae* rDNA silencing during starvation remains to be determined, and whether it is regulated by TORC1.

Remarkably, *S. cerevisiae* histone H2A can be methylated at a glutamine at position 105 (H2AQ105me) by methyltransferase Nop1 and is found to accumulate specifically in the nucleolus. In addition, the ChIP-seq profile indicates that H2AQ105me is enriched within the 35S rDNA regions. Since a mutant with an alanine substitution at Q105 mimicking methylation exhibits increased rRNA transcription, the enrichment in H2AQ105me is assumed to facilitate rRNA transcription [83]. This idea is supported by the finding that the inactivation of TORC1 eliminates H2A methylation from rDNA regions with a concomitant decrease in transcription [84] (Figure 3).

In summary, H3K4 methylation by Set1 and H2AQ105 methylation by Nop1 affect *S. cerevisiae* rRNA transcription negatively and positively, respectively.

### 3.2. Regulatory Mechanisms of RP Gene Expression

The repressive mechanisms of *S. cerevisiae* RP genes are distinct from rDNA and add further complexity. The 137 genes encoding RP (except for 13 genes) carry a UAS (Upstream Activating Sequence), a T-rich element, and a TATA box as upstream regulatory elements. The transcriptional activator Rap1 (repressor/activator protein 1) recognizes the UAS and associates with ~90% of the RP genes [85,86,87,88]. Experiments measuring the expression levels of β-galactosidase from the *lacZ* gene, which is fused downstream of the *RPL16* gene, revealed that transcription levels in cells lacking the Rap1 binding site at the *RPL16* gene promoter were reduced to 10% of that in wild-type cells, confirming that Rap1 activates RP genes [21]. However, when *S. cerevisiae* cells are exposed to heat, osmotic stress, or rapamycin, the localization pattern of Rap1 is not altered despite the observed repression of RP gene transcription, suggesting that Rap1 is just a scaffold for other transcription factors governing their expression [89] (Figure 4).

A systematic study examined the genome-binding regions of 141 *S. cerevisiae* transcription factors identified as sharing DNA-binding and transcriptional activity in the Yeast Proteome Database yielded Fhl1, associated with RP genes, whose function was poorly investigated [90]. A more detailed study using ChIP-on-chip (chromatin immunoprecipitation combined with DNA microarray analysis) showed that Fhl1 accumulates to the promoter of 50–80% of the RP genes [89,91]. Fhl1 works as a transcriptional activator, as the expression of RP genes in cells lacking the *FHL1* gene, was strongly reduced [91]. In addition, Fhl1 fails to bind at eight out of nine RP promoters to which Rap1 also does not bind, supporting the idea that Fhl1 localization depends on Rap1. Moreover, Fhl1 continuously associates with RP genes, as does Rap1, even if cells are exposed to rapamycin, indicating that both Rap1 and Fhl1 are scaffolds for other transcription factors [89] (Figure 4).

The binding sites of transcription factor Ifh1, which has a genetic interaction with those of Fhl1, were also examined by ChIP-on-chip analysis, revealing that both Ifh1 and Fhl1 bind to 46 of RP gene promoters [89,91]. Since overexpression of Ifh1 stimulates RP gene expression, Ifh1 recruited by Rap1 and Fhl1 probably acts as an activator. However, as opposed to Rap1 and Fhl1, Ifh1 is eliminated from almost all RP genes in the stationary phase or by rapamycin treatment [89,91]. Rapamycin treatment abrogates the physical interaction between Fhl1 and Ifh1 [92], which may be regulated by Ifh1 acetylation. Under nutritious conditions, Ifh1 is acetylated by acetyltransferase Gcn5 HAT, whereas rapamycin treatment removes Ifh1 acetylation [93] (Figure 4). This hypothesis could be verified by investigating whether the binding of Fhl1 and Ifh1 is weakened in the *gcn5* deletion strain.

Almost concurrently, Sfp1 was found as a positive regulator of RP gene expression by a screen to isolate factors that change localization under rapamycin and a screen using a deletion library independently [94,95]. Recently, ChIP-exo analysis, capable of detecting protein binding sites at a single nucleotide resolution, has shown that Sfp1 coincides with the Fhl1 and Ifh1 at their binding sites [96]. Since depletion of Ifh1 reduces the association of Sfp1 with RP genes [97], Sfp1 is thought to associate with them via Ifh1 (Figure 4). 

Sfp1 is phosphorylated not only by TORC1 directly but also by PKA. Therefore, Sfp1 is localized in the nucleus under nutrition-rich conditions, whereas starvation or rapamycin treatment promptly evacuates Sfp1 from the nucleus [94,98]. In this way, the accumulation of Ifh1 and Sfp1 activates the transcription of RP genes in nutrient-rich conditions, so how do these factors upregulate gene expression? Reid and colleagues provided a clue to answering this pivotal question. In *S. cerevisiae*, histone H4 acetylation, a marker of transcriptional activity, is catalyzed by the Esa1-containing NuA4 histone acetyltransferase [99]. ChIP combined with a microarray showed that Esa1 accumulates on the RP genes, and this binding is disrupted by rapamycin treatment [100,101]. Since depletion of Esa1 results in a reduction in acetylated H4 enrichment and transcription in RP genes, Esa1 likely upregulates RP gene transcriptions via H4 acetylation [100]. Given that Esa1 fails to associate with RP genes where Rap1 barely binds [100], it is possible that Ifh1 and Sfp1, both of which colocalize with Rap1, recruit Esa1 (Figure 4).

As described above, RP gene expression relies on complicated mechanisms, but the behavior of these factors is dynamically altered when cells shift into nutrient-poor conditions. Starvation and rapamycin treatment triggers the dissociation of Ifh1, Sfp1, and Esa1 from the RP gene. The dissociation of Esa1 was studied by Joo and coworkers [102]. A transcription factor Gcn4 competes with Esa1 for the binding to Rap1, thereby displacing Esa1 from the RP promoter, as demonstrated by in vitro and in vivo experiments [102]. It is expected that Gcn4 competes with Esa1 in a starvation-specific manner by a mechanism in which transcription of *GCN4* is repressed by the general amino acids control (GAAC) pathway during nutritious conditions. In contrast, it is increased during amino acid starvation [103] (Figure 4). 

In addition to the dissociation of Esa1, the downregulation of RP genes is enhanced by histone deacetylase Rpd3. While several studies reported that Rpd3 constitutively localizes to the RP genes [101,104], Humphery et al. suggested that Rpd3 only binds after rapamycin treatment [105]. In a seeming paradox, one may want to consider the binding of the transcriptional repressors Stb3 and Crf1 to these promoters [106]. Stb3, phosphorylated by Sch9, which acts downstream of TORC1, localizes to the cytoplasm, whereas it migrates to the nucleus after starvation or rapamycin treatment and accumulates at the RP gene promoters [106,107]. Rpd3 is then recruited onto RP genes through a scaffold of Stb3 [105] (Figure 4). Crf1 was identified by a two-hybrid screen in search of factors that bind to Fhl1. Since the transcriptional repression of RP genes by rapamycin treatment is alleviated in the *crf1*∆ mutant, Crf1 is considered a repressor of RP genes during starvation. Crf1 resides in the cytoplasm in nutritious conditions, whereas it is phosphorylated by the PKA-regulated kinase Yak1, which becomes activated upon starvation, migrates to the nucleus, and binds to Fhl1 [92]. Therefore, Crf1 might recruit Rpd3 to the RP genes in concert with Stb3 (Figure 4). Enigmatically, Crf1 functions as an RP gene repressor in the TB50 strain but not in the W303 strain background [108]. Therefore, it is plausible that the transcriptional repression of RP genes mainly relies on the recruitment of Rpd3 via Stb3.

In summary, the transcription of RP genes is strictly regulated by the dynamic localization of transcription factors and histone modification enzymes before and after starvation.

### 3.3. Regulatory Mechanisms of Ribi Gene Expression

In addition to the rRNAs and RPs that compose the ribosome, more than 200 other factors (Ribi) are involved in *S. cerevisiae* ribosome biogenesis [43]. The upstream sequence of the Ribi gene consists of RRPE and PAC sequences distinct from that of RP genes [109,110,111,112]. Therefore, Ribi gene transcription is regulated by a partially distinct mechanism from the RP genes. In a comprehensive analysis of proteins that bind to the RRPE sequence, Stb3 was identified as associating with RRPE during in vitro and in vivo experiments [113]. It is known that Stb3 contributes to the transcriptional repression of Ribi genes based on the finding that overexpression of Stb3 reduces their expression [107]. Additionally, Stb3 is sequestered to the cytoplasm by phosphorylation through the TORC1 pathway during nutrient-rich conditions. In contrast, upon starvation, it accumulates at the RRPE motif of Ribi gene promoters and represses their transcription [107] (Figure 5).

In a comprehensive analysis of DNA sequence motifs that bind transcriptional factors, Dot6 and Tod6 were found to bind to the PAC motif [114,115,116]. *S. cerevisiae* cells lacking both *DOT6* and *TOD6* genes exhibited mitigation of Ribi gene repression during starvation, suggesting that Dot6 and Tod6 downregulate their expression [117]. Moreover, a systematic analysis using mass spectrometry that detects proteins phosphorylated by TORC1 and its downstream factors, Sch9 and Tap42, revealed that Dot6 and Tod6 are phosphorylated by this pathway [75]. Huber and colleagues further indicated that R[R/K]x[S/T]* motifs of Dot6 and Tod6 are preferentially phosphorylated by Sch9 and PKA in vivo [106]. Dephosphorylation of Dot6 and Tod6 upon starvation and rapamycin treatment trigger the translocation of its localization from the cytoplasm to the nucleus, where they bind the Ribi gene promoters [118]. Taken together, Stb3 binds to the RRPE sequence and Dot6/Tod6 to the PAC sequence during starvation, respectively. These factors promote histone deacetylation by recruiting Rpd3 and repressing Ribi genes (Figure 5).

Then, how is the transcriptional activity of Ribi genes accomplished in nutritious environments? Although one report claimed that Sfp1, which is associated with RP genes, does not bind to Ribi gene promoters [96], ChEC-seq (endogenous chromatin cleavage-sequencing) analysis has recently revealed that Sfp1 accumulates on Ribi genes [97]. Importantly, Ihf1 hardly binds to Ribi gene promoters, although Sfp1 associates via a scaffold with Ifh1. This suggests that Sfp1 directly interacts with the Ribi gene sequence independent of Ifh1 [97]. It is, however, unclear whether Sfp1 recruits Esa1 since Rap1 is fundamentally absent on Ribi genes. Nevertheless, given an intriguing report that the depletion of Esa1 resulted in a decrease in both H4 acetylation and RNA polymerase II at most Ribi gene promoters, Esa1 is likely to accumulate on the Ribi genes [119] (Figure 5).

In summary, as shown in Figure 5, Ribi gene expression may be partially regulated by a common mechanism with RP gene regulation, while both can independently function.

## 4. Comparison of Regulatory Factors Repressing Ribosome-Related Genes between *S. cerevisiae* and *S. pombe*

In *S. pombe*, the molecular mechanisms underlying transcriptional activation during starvation have been extensively studied, mainly focusing on the gluconeogenic *fbp1* gene [120,121]. However, research on *S. pombe* regarding the mechanism by which ribosome-associated genes are transcriptionally repressed is substantially lagging behind that of *S. cerevisiae*. Hereafter, we summarize the *S. pombe* homologs of transcription factors contributing to the ribosomal gene repression in *S. cerevisiae*.

Hmo1, which widely localizes on rDNA in *S. cerevisiae*, is also suggested to promote rDNA transcription in *S. pombe* [122]. However, the behavior of Hmo1 under starvation and whether it is regulated by TORC1 is unknown.

A homolog of the transcriptional activator Sfp1 is also present in *S. pombe*; however, it is currently unclear whether Sfp1 regulates ribosome-related gene transcription. Recently, a cap analysis of gene expression (CAGE) demonstrated that the Sfp1-binding motif is concentrated in the promoter of ribosome biosynthetic genes repressed during nitrogen starvation [123], indicating that Sfp1 may somehow contribute to the *S. pombe* ribosomal gene expression. 

The forkhead protein Fhl1 is also regulated by TORC1 in *S. pombe*. However, Fhl1 predominantly contributes to the regulation of genes related to the transport process and mating/sporulation, with little or no influence on the ribosome-related genes [124].

Although *S. cerevisiae* Crf1 and Ifh1 are annotated with the *S. pombe* homolog Crf1, it has not been determined whether Crf1 regulates ribosome-related genes. A homolog of Stb3 is also present in *S. pombe*, but its function has yet to be discovered. Finally, there is no annotation and homolog information for Dot6 and Tod6 on PomBase (https://www.pombase.org/ accessed on 31 January 2023). Altogether, the analysis of transcription factors equivalent to those revealed in *S. cerevisiae* is poor in *S. pombe*.

## 5. Epigenetic Regulatory Mechanisms of rDNA Regions in *S. pombe*

On the contrary, several studies have shown that some histone deacetylases contribute to *S. pombe* ribosomal gene repressions. Yet, histone deacetylase (class I) Clr6, homologs of Rpd3 repressing rRNA, RP, and Ribi genes in *S. cerevisiae*, do not contribute to the repression of ribosome-related genes. In contrast, class II deacetylase Clr3, a homolog of Hda1, represses at least rRNA gene expression [125,126]. Moreover, Sir2 and Hst2, among the three Sirtuin families in *S. pombe*, accumulate at rDNA regions to repress transcription [127]. These results were obtained under nutritious conditions, suggesting that part of the rDNA repeats is inactivated in *S. pombe* as in *S. cerevisiae* (Figure 6).

One significant discrepancy regarding the transcriptional repression mechanism between budding yeast and fission yeast is the heterochromatin formation machinery, which is only present in fission yeast [19]. Indeed, heterochromatinization by the H3K9 methyltransferase Clr4 and the heterochromatin proteins Swi6 and Chp2 (HP1 in humans) in concert with the RNAi-dependent pathway has been found in rDNA regions [128]. Such a heterochromatin formation is already observed in nutrient-rich environments and may prevent mitotic recombination between rDNA repeats, in addition to inactivating part of the rDNA repeats [129,130,131,132] (Figure 6).

In the actively transcribing rDNA repeats, ATF/CREB family transcription factor Atf1 (ATF-2 in humans) positively regulates rRNA gene expression [132]. Previous studies showed that stress-responsive transcription factor Atf1 accumulates at the promoter of the core environmental stress response (CESR) genes to activate the downstream gene in response to environmental cues, such as starvation [29,133,134,135]. On the other hand, it was reported that Atf1 directly binds to the histone methyltransferase Clr4 and silences the mating-type loci [136,137]. With this background, we conducted ChIP-seq for Atf1 binding sites before and after starvation, revealing that Atf1 broadly localizes on rDNA during nutrient-rich conditions but is dissociated upon starvation. Disruption of the *atf1* gene caused an enhancement in H3K9 methylation and repression of rRNA genes, suggesting that Atf1 works as a transcriptional activator [132]. Additionally, the dissociation of Atf1 upon starvation triggers the accumulation of histone chaperone FACT [132], which maintains methylated histones by preventing histone turnover [138] (Figure 6). Since FACT potentially recognizes the Q105 site in H2A [83,139,140], it would be intriguing to study whether Atf1 regulates the modification of H2AQ105 in response to nutrient limitation.

In addition to, and independent of Atf1, acetyltransferase Gcn5 HAT accumulates in rDNA regions and dissociates upon starvation [132]. Inactivation of Gcn5 only in the nucleolus caused increased H3K9 methylation in rDNA chromatin, suggesting that Gcn5 competes with deacetylase and methyltransferase to prevent heterochromatin formation in the actively transcribing rDNA repeats [132]. Since Gcn5 is dissociated from rDNA upon starvation, histone H3 retained in the actively transcribing rDNA repeats is predominantly methylated by the RNAi pathway (Figure 6). Since Oya et al. showed that ubiquitination of H3K14 by CLRC complex is required for H3K9 methylation [141], H3K14 deacetylation by Clr3 is a prerequisite for H3K14 ubiquitination, leading to H3K9 methylation. Moreover, it is noteworthy that nitrogen starvation leads to the degradation of exosome/TRAMP, resulting in the enhancement in Ago1-associated sRNAs triggering heterochromatinization by the RNAi-dependent pathway on rDNA regions [17,142] (Figure 6).

As described above, transcription from rDNA in *S. pombe* is repressed by mechanisms quite different from those in *S. cerevisiae*. However, whether TORC1 and PKA participate in these mechanisms is unclear, and the same is true for whether mechanisms uncovered in rDNA also regulate ribosome-related genes. Intriguingly, H3K9 methylation in ribosome-related genes increased when *S. pombe* cells were exposed to nitrogen starvation since the sRNAs derived from transcriptions of ribosome-related genes were enhanced by Ago1, leading to the activation of the RNAi-dependent pathway [17]. Additionally, we found that the transcriptional activator Atf1 dissociated from ribosome-related genes upon glucose starvation, and transcription of *rpl102*, *rlp7*, and *gar2* genes was decreased in *atf1*∆ cells, suggesting that the dissociation of Atf1 causes transcriptional repression of ribosome-related genes [132].

## 6. Perspectives

Mutations that cause complete loss of ribosomal function result in lethality in yeast and embryonic lethality in higher eukaryotes. Alternatively, partial loss of ribosomal function causes a variety of maladies called ribosomopathy in humans [143]. The etiology of Diamond–Blackfan anemia, a representative ribosomal disease, is a mutation of ribosomal genes [144,145]. Other diseases such as isolated congenital asplenia [146], childhood cirrhosis [147], chromosome 5q- syndrome [148], Treacher Collins syndrome [149], and Shwachman–Bodian–Diamond syndrome [150] are also caused by mutations in the ribosomal gene. Among these diseases, the etiology of Treacher Collins syndrome is a mutation in the *TCOF1* gene, encoding the cofactor Treacle of RNA polymerase I, which transcribes rRNA [151]. This finding suggests that disrupting the rigorous regulation of rRNA expression levels may be the etiopathogenic factor. However, there are currently no findings on the transcription factors discussed in this paper regarding their role as etiologic agents of ribosomopathies. 

In HEK293 cells and β-TC6 cells, a mouse pancreatic beta cell, mTOR binds to the rDNA promoter, and this binding is dissociated upon rapamycin treatments and amino acid starvation [152]. This suggests that the behavior of mTOR is equivalent, at least in *S. cerevisiae*. Additionally, methylated histone H3 and HP1 are accumulated in rDNA regions in NIH3T3 cells. NoRC, a nucleolar remodeling complex, mediates this, but it remains unknown whether another factor, such as ATF-2 (a homolog of Atf1 in *S. pombe*), also contributes [153]. In the future, we expect that defects in transcription factors that regulate the expression of ribosome-associated genes and cause ribosomopathies will be studied using *D. melanogaster* or mice as model organisms.

## Figures and Tables

**Figure 1 biomolecules-13-00288-f001:**
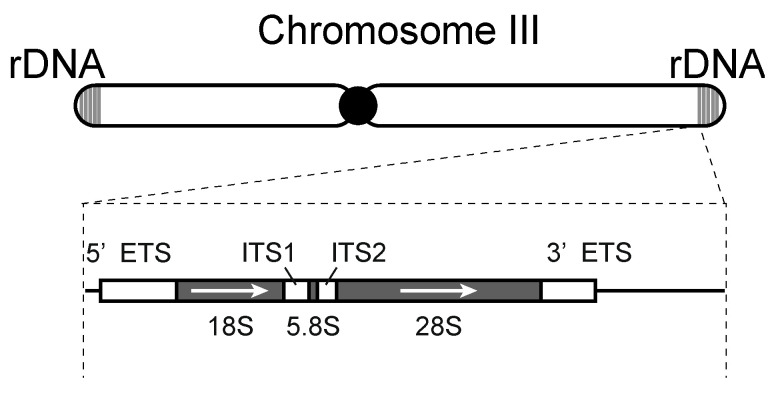
The structure of rDNA in *S. pombe*. rDNA regions exist on both ends of chromosome III in *S. pombe*. A single repeat of rDNA is shown in the dotted black box. 18S, 5.8S, and 28S rRNAs are transcribed by RNA polymerase I as the precursor rRNA from 5′ETS (external transcribed spacer) to 3′ETS. Four transcribed spacers (white boxes), including ITS (internal transcribed spacer) 1 and ITS 2, are removed by endonuclease and exonuclease processing.

**Figure 2 biomolecules-13-00288-f002:**
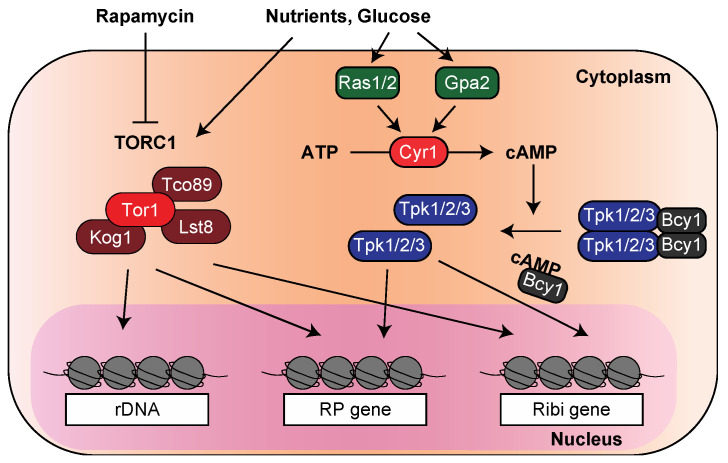
Regulatory mechanism of ribosomal genes by TOR and PKA pathways. Of the two TOR complexes, TORC1, composed of Tor1, Kog1, Lst8, and Tco89, primarily regulates the transcription of ribosomal genes. Nutrient limitation or rapamycin treatments causes the inactivation of TORC1, leading to the downregulation of ribosomal gene expressions. The PKA pathway also contributes to the expression of ribosomal genes. In the presence of glucose, Cyr1, activated by Ras1/2 and Gpa2, converts ATP to cAMP. As a result of cAMP inhibiting Bcy1, free Tpk activates the transcription of ribosomal genes.

**Figure 3 biomolecules-13-00288-f003:**
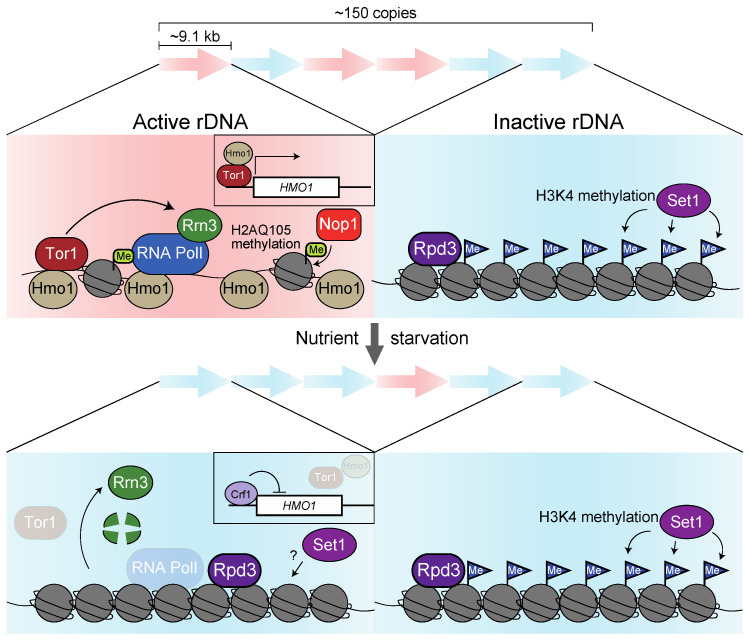
Regulatory mechanisms of rRNA expression in *S. cerevisiae*. A graphical view of how rDNA genes are transcriptionally regulated before and after starvation. In *S. cerevisiae*, half of the 150 copies of rDNA genes are active (the pink area). In contrast, nutrient starvation inactivates many repeats by the degradation of Rrn3, dissociation of Hmo1, and loss of H2AQ105 methylation (the blue area). Upon starvation, TORC1 deprivation leads to the accumulation of repressor Crf1 at the *HMO1* promoter, resulting in the downregulation of *HMO1* expression (in black square boxes). The deacetylase Rpd3 maintains a closed-chromatin formation, and H3K4 methyltransferase Set1 represses rRNA transcription in inactive rDNA repeats.

**Figure 4 biomolecules-13-00288-f004:**
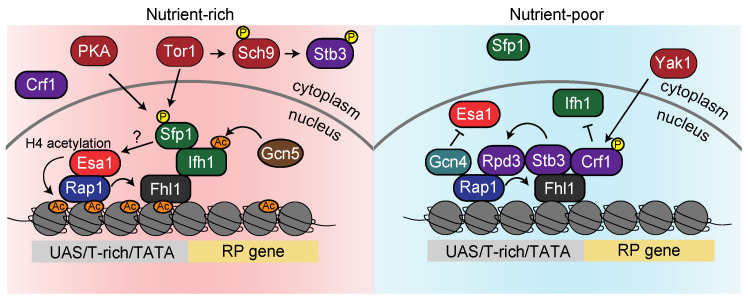
Regulatory mechanisms of RP gene expression in *S. cerevisiae*. In nutrient-rich environments (the pink area on the left), histone acetyltransferase Esa1 is recruited to the promoter of RP genes via interaction with Rap1. Ifh1 and Sfp1, which are recruited via the Fhl1 scaffold, also contribute to the expression of RP genes. In addition, the repressor Stb3 is phosphorylated by Sch9, which acts downstream of TORC1 (Tor1), preventing its accumulation at RP gene promoters. In nutrient-poor environments (the blue area on the right), Gcn4, interacting with Rap1, prevents the recruitment of Esa1 to RP genes. Moreover, Ifh1 and Sfp1 are dissociated from Fhl1, and the repressors Crf1 and Stb3 are recruited instead, leading to the accumulation of Rpd3.

**Figure 5 biomolecules-13-00288-f005:**
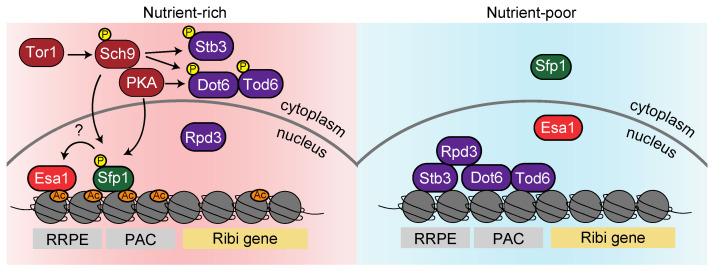
Regulatory mechanisms of Ribi gene expression in *S. cerevisiae*. In nutrient-rich conditions (the pink area on the left), Sfp1 resides at Ribi gene promoters without Ifh1. Sfp1 might recruit Esa1 and upregulate the transcription of Ribi genes. Stb3, Dot6, and Tod6 phosphorylated by Sch9 and PKA are in the cytoplasm. During nutrient starvation (the blue area on the right), Stb3 binds to the RRPE motif (AAAAATTT) of Ribi genes, and Dot6/Tod6 accumulates at the PAC motif (CTCATCG). As a result of the recruitment of Rpd3 by these factors, Ribi gene expression is repressed.

**Figure 6 biomolecules-13-00288-f006:**
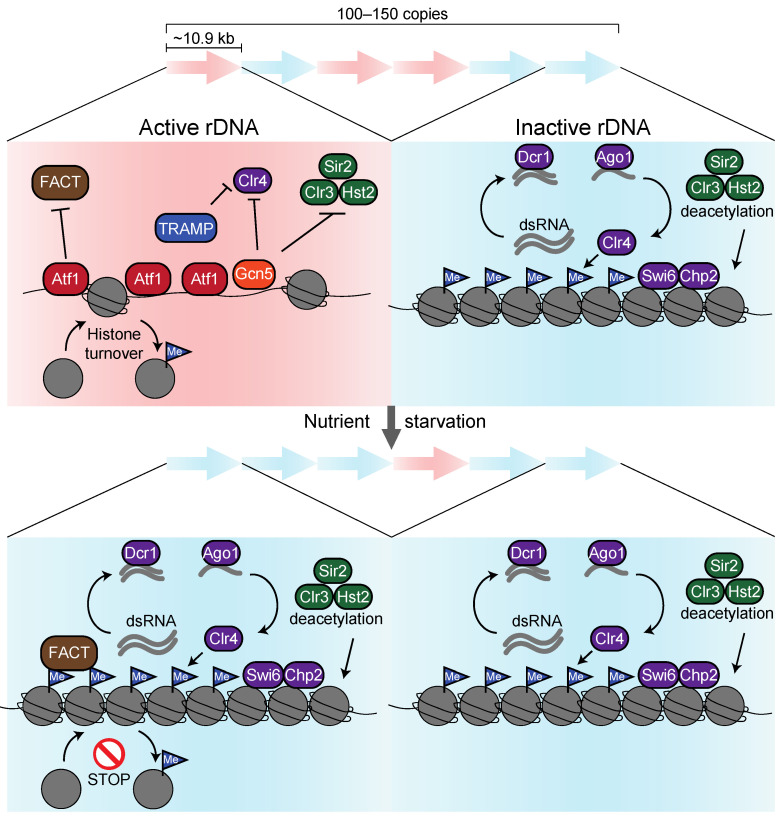
Regulatory mechanisms of rDNA gene expression in *S. pombe*. In nutrient-rich conditions (the pink area on the left), transcription factor Atf1 prevents the accumulation of histone chaperone FACT. Gcn5 HAT competes with histone deacetylase and methyltransferase to inhibit hypermethylation of H3K9. Without FACT, histone turnover is facilitated so that H3K9 methylation is quickly removed. TRAMP/exosome prevents the deposition of RNAi-related proteins in rDNA, preventing H3K9 methylation and heterochromatinization. Disruption of RNAi-related genes or deacetylase genes causes an enhancement in rRNA expression, suggesting that an RNAi-dependent pathway accelerates heterochromatin formation in dormant rDNA repeats. In nutrient starvation (the blue area on the right), Atf1 and Gcn5 dissociate from rDNA, leading to the recruitment of FACT. Methylated histones are maintained by FACT, which prevents histone turnover. Removal of TRAMP/exosome from rDNA accelerates Ago1-dependent sRNA generation, resulting in the heterochromatin formation by RNAi-dependent pathway.

## Data Availability

Not applicable.

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
