# Peer review of "Comparative Research: Regulatory Mechanisms of Ribosomal Gene Transcription in Saccharomyces cerevisiae and Schizosaccharomyces pombe"

_biomolecules, 2023, doi:10.3390/biom13020288_

Round 1
Reviewer 1 Report
The manuscript at hand gives a nice overview on how ribosome production is regulated both in budding and fission yeast at the level of gene expression. It tries to compare the two yeast species, which is challenging, regarding the lack of knowledge still prevalent for S. pombe. Yet, the authors to a good job with summarizing the data available.
There is only one small criticism regarding the regulatory schemes: whenever necessary, but especially in Figures 3 and 4, I would suggest to include the nuclear membrane to make clear, which factors depend on their translocation between cytosol and nucleus for their regulation. In addition, in Figure 2 I would suggest not to imply a total repression of the rDNA under starvation conditions by drawing all arrows in blue in the lower part. Although there may be considerable reduction of gene expression, it is surely not completely abolished.
As for the English and some other queries, I have extensively marked up the manuscript with suggestions and will attach the PDF file, rather than copying all comments, here.

Reviewer 2 Report
In the manuscript, biomolecules-2177170, Hirai and Ohta summarized our current understanding of the regulatory mechanisms of ribosomal gene transcription in in Saccharomyces cerevisiae and Schizosac-3 charomyces pombe.
Specific concerns:
1. The organization of the main text can be improved. It seems that Sections 2 and 3 can be merged into Section 5, highlighting the differences in the transcriptional regulation of the ribosomal genes in stress response between the two species.
2. With respect to the regulatory mechanisms of ribosome-related gene expression in budding yeast, (1) is the presence of the UAS/T-rich/TATA motifs in the promoter specific to the 137 RP genes, or are these elements also existed in other protein-coding genes transcribed by RNA PolII in the yeast cells? i.e., are the mechanisms observed unique to the 137 RP genes but not other protein-coding genes? (2) Are the RRPE and PAC motifs only existed in the Ribi genes? It will be helpful to include the sequences of RRPE and PAC in the legend of Figure 4.
3. It seems there are very limited insights into the regulatory mechanisms of ribosome-related gene expression in fission yeast.
4. In the perspectives, the authors mentioned a few human ribosomopathy diseases. However, due to the distinct mechanisms underlying ribosome gene regulation between mammalians and yeasts, the link between the human diseases and the regulatory mechanisms in yeast seems weak.
Reviewer 3 Report
This manuscript represents a fair introduction to the regulation of ribosome biogenesis in budding and fission yeast. However, at this stage, the manuscript needs to provide more details concerning the molecular mechanisms that differentiate between the two types of yeast.
1. The discussion of the “Regulation of ribosome-associated gene expression by TOR and PKA pathways” is incomplete.
a. In this same section, the authors refer to ribosomal genes? What are ribosomal genes?
b. This section would benefit from a figure illustrating the pathways and the differences.
2. Line 143. The authors write, “A region of the rDNA gene repeated 150 times of ~9.1kb unit is present on chromo-143 some XII in S. cerevisiae”. Actually, the rDNA gene (transcribed and nontranscribed spacer represents a unit that is repeated.
3. Figure 2 is vastly oversimplified. The authors need to consider open ribosomal chromatin that is not transcribed.
4. Line 168. I am not aware that Pol I itself can remove histones.
5. Line 177. The authors need to refer to the Figure and the Figure legend. Further, the authors appear to ignore the finding that in yeast, if not mammals, TOR regulates the activity of Rrn3.
6. Line 180 TORC1 dysfunction?
7. Line 181. It is not clear that the response to nutrient deprivation results in the rapid depletion of Hmo1 such that it would alter rDNA transcription. Nor is it clear that the " rapid" repression of rDNA transcription results from Hmo1-dependent alteration of the rDNA chromatin.
8. Line 188. Tsochner's laboratory reported the TOR-dependent reduction in the expression level of Rrn3p lowers the activity of the yeast RNA Pol I machinery, but did not account for the strong inhibition of rRNA synthesis. Further, others have reported that the inhibition of TOR results in the inhibition of phosphorylation of Rrn3 that leads to its inactivation.
9. The CARA construct is the fusion of Rrn3 to rpa43, hence the i9nteraction of Rrn3 with Pol I, that is mediated by rpa43, is independent of the phosphorylation status of Rrn3.
10. Not “reversely”, “conversely”.
11. Line 213. Please explain/define the “expansion” promoter. Also, Also , there is significant discussion in the literature of the roles of the sirtuins in the modification of Pol I (A49). The authors need to say more about this.
12. Line 215. Are the authors suggesting that COMPASS represses the fraction of ribosomal genes that are normally active? Or is the complex essential for maintaining the status quo of the inactive ribosomal RNA chromatin?
13. Line 234. This sentence needs to be rewritten/clarified. “In cells lacking the sequence upstream of the RPL16 234 gene encoding a ribosomal protein, its transcriptions are reduced to 1/10, suggesting that Rap1 activates RP genes.”
14. Line 251. The ChIP-on-ChIP experiment alluded to needs to be described. C-on-C would not be necessary for the experiment described.
15. Line 261. Do the two proteins interact or do they compete with one another for binding? The Saccharomyces genome data base says, "Coactivator, regulates transcription of ribosomal protein (RP) genes; recruited to RP gene promoters during optimal growth conditions via Fhl1p; subunit of CURI, a complex that coordinates RP production and pre-rRNA processing; regulated by acetylation and phosphorylation at different growth states via TORC1 signaling; IFH1 has a paralog, CRF1, that arose from the whole genome duplication"
16. Line 295. This sentence needs to be stated more clearly. ‘Gcn4, a transcription factor with a bZIP that binds to the consensus sequence 293 ATGA(C/G)TCAT, competes with Esa1 to bind Rap1, thereby displacing Esa1 from the 294 RP promoter, as demonstrated by in vitro and in vivo experiments.
17. Line 310…”was” not “is”.
18. Line 312. IS Crf1 a substrate for TOR? Further, this model should be presented with a complementary figure.
19. Line 326. PAC was identified in promoters of genes transcribed by Pol II that encode Pol I and Pol III subunits.. Also, the authors should present the RPE (ribosomal RNA processing element sequence (AAAAATTT) and the Polymerase A and C element (CTCATCG).
